# CT-Based Radiomics Helps to Predict Residual Lung Lesions in COVID-19 Patients at Three Months after Discharge

**DOI:** 10.3390/diagnostics11101814

**Published:** 2021-09-30

**Authors:** Jia Huang, Feihong Wu, Leqing Chen, Jie Yu, Wengang Sun, Zhuang Nie, Huan Liu, Fan Yang, Chuansheng Zheng

**Affiliations:** 1Department of Radiology, Union Hospital, Tongji Medical College, Huazhong University of Science and Technology, 1277 Jiefang Avenue, Wuhan 430022, China; huangjiapal@hust.edu.cn (J.H.); wfh_wuhan@hust.edu.cn (F.W.); leqingchen@hust.edu.cn (L.C.); gavinyuwhuh@hust.edu.cn (J.Y.); swg_xh@hust.edu.cn (W.S.); niezhuang@hust.edu.cn (Z.N.); 2Hubei Province Key Laboratory of Molecular Imaging, Wuhan 430022, China; 3Precision Healthcare Institute, GE Healthcare, Shanghai 201203, China; Huan.Liu@ge.com

**Keywords:** COVID-19, radiomics, tomography, X-ray computed, residual lesion

## Abstract

Background: In this study, our focus was on pulmonary sequelae of coronavirus disease 2019 (COVID-19). We aimed to develop and validate CT-based radiomic models for predicting the presence of residual lung lesions in COVID-19 survivors at three months after discharge. Methods: We retrospectively enrolled 162 COVID-19 confirmed patients in our hospital (84 patients with residual lung lesions and 78 patients without residual lung lesions, at three months after discharge). The patients were all randomly allocated to a training set (*n* = 114) or a test set (*n* = 48). Radiomic features were extracted from chest CT images in different regions (entire lung or lesion) and at different time points (at hospital admission or at discharge) to build different models, sequentially, or in combination, as follows: (1) Lesion_A model (based on the lesion region at admission CT); (2) Lesion_D model (based on the lesion region at discharge CT); (3) Δlesion model (based on the lesion region at admission CT and discharge CT); (4) Lung_A model (based on the lung region at admission CT); (5) Lung_D model (based on the lung region at discharge CT); (6) Δlung model (based on the lung region at admission CT and discharge CT). The area under the receiver operating characteristic curve (AUC), sensitivity, and specificity were used to evaluate the predictive performances of the radiomic models. Results: Among the six models, the Lesion_D and the Δlesion models achieved better predictive efficacy, with AUCs of 0.907 and 0.927, sensitivity of 0.898 and 0.763, and specificity of 0.855 and 0.964 in the training set, and AUCs of 0.875 and 0.837, sensitivity of 0.920 and 0.680, and specificity of 0.826 and 0.913 in the test set, respectively. Conclusions: The CT-based radiomic models showed good predictive effects on the presence of residual lung lesions in COVID-19 survivors at three months after discharge, which may help doctors to plan follow-up work and to reduce the psychological burden of COVID-19 survivors.

## 1. Introduction

The pandemic of the coronavirus disease 2019 (COVID-19) has brought significant health and economic losses around the world. More than 100 million people worldwide have recovered from COVID-19 so far. In the post-COVID-19 stage, the epidemiology, clinical features, pathogenesis, and complications of COVID-19 have been clearly described [1,2], but the long-term sequelae caused by this infectious disease are still unknown to some extent.

The lungs are the main target organ of COVID-19 and present typical radiographic signs of pneumonia in the acute phase, which include bilateral patchy shadows or ground glass opacity [3,4]. Survivors of COVID-19 are frequently reported to have pulmonary sequelae, such as dyspnea, residual lung lesions, and lung function impairment [5,6,7]. Studies have found that more than half of the patients discharged had residual lung lesions on CT scans [8,9], which may cause patient distress. Therefore, early prediction of residual lung lesions in COVID-19 patients is important for planning follow-up work and reducing the psychological burden of survivors.

Chest computed tomography (CT) is widely used for the screening, diagnosis, and follow-up of COVID-19 because of its characteristics of timeliness, rapidity, high positive rate, and close correlation between the scope of lung lesions and clinical symptoms [10]. Radiomics is a new method, proposed in recent years, that combines big data technology and medical image-assisted diagnosis, which can provide valuable information for the diagnosis, monitoring, and prognostic evaluation of diseases [11]. However, most CT-based radiomic studies of COVID-19 have focused on distinguishing COVID-19 from other types of pneumonia, predicting the clinical types and severity, and predicting the risk of death and intensive care unit (ICU) admission. To date, no radiomic studies have predicted the recovery of COVID-19 survivors after discharge from hospital.

Therefore, the aim of our study is to develop and validate a CT-based radiomic model predicting the presence of residual lung lesions in COVID-19 survivors at three months after discharge.

## 2. Materials and Methods

### 2.1. Patients

Our retrospective study was approved by the Institutional Ethics Review Board of Union Hospital, Tongji Medical College, Huazhong University of Science and Technology (no. 0044), and informed consents were waived. A total of 187 adult patients with virus real-time reverse transcription polymerase chain reaction test (RT-PCR)-confirmed COVID-19 pneumonia who were hospitalized in our hospital from 6 January 2020 to 14 April 2020 and followed up from 1 May 2020 to 31 July 2020 were included in this study. Twenty-five patients were subsequently excluded due to chest CTs showing no lung lesions in COVID-19 patients at admission or discharge. Discharge standards referred to the Diagnosis and Treatment Protocol for COVID-19 in China (Trial Fifth Edition) [12]. Finally, 162 patients with residual lung lesions at discharge and follow-up CTs at three months after discharge were enrolled in the study.

### 2.2. CT Image Acquisition and Segmentation

The enrolled patients all underwent chest CT examinations at admission, at discharge, and at three months after discharge. The images were all acquired with patients lying in a supine position, at the end of inspiration, and without contrast agent administration. All scans were performed on four CT scanners randomly: one 16-detector row scanner, Toshiba Medical Systems (Otawara, Japan) and three 64-detector-row scanners, Siemens Healthineers (Erlangen, Germany), GE Healthcare (Waukesha, WI, USA), and Philips Healthcare (Best, the Netherlands). The CT acquisition parameters included a tube voltage of 120 or 100 KVp and adaptive tube current. The images were reconstructed at a slice thickness of 1.5 mm for the GE, Siemens, and Philips Healthcare scanners, and 2.0 mm for the TOSHIBA scanner.

On the CT images of the patients at the time of admission and discharge, we used the uAI Intelligent Assistant Analysis System (United Imaging Medical Technology Company Limited, Shanghai, China) [13] to pre segment both lungs (entire lung volume) and lesions caused by COVID-19. Subsequently, two radiologists (J.H. and F.H.W., both with 5 years of experience) manually corrected the results of the pre-segmentation for all 162 patients in consensus and derived the volumes of interest (VOIs).

### 2.3. Subjective Assessment of Follow-Up CTs 

According to the results of the uAI Intelligent Assistant Analysis System, two radiologists (J.H. and F.H.W.) evaluated the residual lung lesions with consensus on the follow-up CT in lung window (at a modifiable window width of 1200 HU and window level of −600 HU) after fully comparing the admission, discharge, and follow-up CTs of each patient. Discussion and consensus resolved all differences. Subsequently, another experienced radiologist (F.Y., with 27 years of experience) made the final confirmation. According to the final decision of the radiologists, the patients were divided into a group with residual lung lesions (RLL) or a group without residual lung lesions (NRLL).

### 2.4. Radiomic Feature Extraction

The workflow of radiomic research is shown in Figure 1. Before the feature extraction, isotropic voxel resampling into 1 × 1 × 1 mm with linear interpolation was used for image preprocessing to normalize the geometry of the CT images. The corrected VOIs were imported to extract radiomic features using LK software (Lung Kit, AK, Version V2.3.0.R, GE Healthcare). A total of 1218 radiomic features were extracted and classified into seven groups: first order, shape, gray level co-occurrence matrix (GLCM), gray level size-zone matrix (GLSZM), gray level run-length matrix (GLRLM), neighborhood gray tone difference matrix (NGTDM), and neighboring gray level dependence matrix (GLDM). To enhance intricate patterns in the data that could be invisible to the human eye, advanced filters were applied which included: Laplacian of Gaussian (LoG, sigma 2.0 and 3.0 mm) and wavelet decompositions with all possible combinations of high (H) or low (L) pass filter in each of the three dimensions (HHH, HHL, HLH, LHH, LLL, LLH, LHL, and HLL).

### 2.5. Radiomic Feature Selection and Model Construction

A total of 1218 radiomic features were automatically extracted for each segmented VOI (Lesion_A, Lesion_D, Δlesion, Lung_A, Lung_D, and Δlung). Before feature selection, the abnormal or missing values were replaced by the median, and feature standardization was applied. The maximum relevance and minimum redundancy (mRMR) method was used to exclude the redundant features and kept the most relevant features with targets. Least absolute shrinkage and selection operator (LASSO) was used to reduce the redundancy or selection bias of the features. AIC was used to measure the goodness of model fitting and model complexity. The radiomic scores were calculated by multiplying selected features and their corresponding coefficients.

The 162 patients were grouped into a training set (*n* = 114) and a test set (*n* = 48) using a stratified random resampling method. Machine learning algorithms were applied to construct radiomic models predicting the presence of residual lung lesions. According to the differences among VOIs, we established six radiomic models. The Lesion_A model extracted radiomic features of lesions from the admission CT, while the Lesion_D model extracted radiomic features from the discharge CT. The Lung_A model extracted radiomic features of the total lung from the admission CT, while the Lung_D model extracted radiomic features from the discharge CT. ΔFeatures were defined as the percentage change in radiomic features from discharge CT to admission CT, which provided information on the evolution of feature values [14,15]. The Δlesion and Δlung models were derived from the following formulas, respectively:Δlesion = (Lesion_D-Lesion_A)/Lesion_A,
Δlung = (Lung_D-Lung_A)/Lung_A.

### 2.6. Statistical Analysis

The statistical analysis was performed using the Institute of Precision Medicine Statistics (IPMs, version 2.1, GE Healthcare) and SPSS 26.0 software (IBM Corp, Armonk, NY, USA). Categorical variables were expressed as counts and percentage, while continuous variables were expressed as medians (25th percentile and 75th percentile). The differences among all the variables between the RLL and NRLL groups were assessed using the Mann–Whitney U test for continuous variables, and the chi-square test or Fisher’s exact test for categorical variables. The area under the receiver operating characteristic (ROC) curve (AUC), sensitivity, and specificity were used to evaluate the predictive performances of the models. The optimal cut-offs to predict the presence of residual lung lesions were identified by Youden’s index. The AUCs of different models on different datasets were compared using the Delong test. *p*-values of <0.05 were considered to be statistically significant.

## 3. Results

### 3.1. Patient Characteristics

The 162 patients (84 patients with residual lung lesions and 78 patients without residual lung lesions) included 65 (40.12%) males and 97 (59.88%) females. The median age of the 162 patients was 56.00 (43.00, 63.25) years, and the median length of hospital stay was 20.00 (13.00, 28.25) days. The interval from discharge date to follow-up CT was 103 (83, 124) days. The flow diagram for patient selection is shown in Figure 2.

The baseline characteristics of patients in the RLL group and the NRLL group are shown in Table 1. In both the training set and test set, patients in the RLL group were older than those in the NRLL group. In the test set, there were significant differences in the gender distribution of the patients and the length of hospital stay. There was no statistical difference in the time interval from discharge to follow-up CT between the RLL group and the NRLL group.

### 3.2. Establishment of Radiomic Signature

Before feature selection, the abnormal or missing values were replaced by the median, and feature standardization was applied. Next, the mRMR and LASSO were used to select the most optimal features. After the redundant and irrelevant features were removed by mRMR, 70 features from each VOI were retained. Then, the LASSO was conducted to decrease the feature redundancy with the Akaike criterion. The LASSO includes choosing the regular parameter λ and determining the number of the features. After the number of features was determined, the most predictive subset of features was chosen and the corresponding coefficients were calculated. 

The selected features and their corresponding coefficients of the six radiomic models are shown in Table 2. After radiomic feature selection, 6 features on Lesion_A model, 8 features on Lesion_D model, 13 features on Δlesion model, 3 features on Lung_A model, 7 features on Lung_D model, and 5 features on Δlung model were finally chosen in the training set. These features are significantly different between RLL and NRLL groups (all *p* < 0.05). Furthermore, there are three radiomic features repeated in the six models: log.sigma.3.0.mm.3D_glcm_InverseVariance, wavelet.HHL_glcm_ClusterShade, and log.sigma.3.0.mm.3D_firstorder_Skewness. The selected features and their corresponding coefficients establish the radiomic signature, and the radiomic scores for each patient in the different models are displayed in a bar chart (Figure 3).

### 3.3. Evaluation of Model Performance

The predictive efficacy of each model is shown in Table 3, and the results of the ROC curve analysis are shown in Figure 4. In the training set, the Lesion_D and Δlesion models achieved higher predictive efficacy with AUCs of 0.907 and 0.927, sensitivity of 0.898 and 0.763, and specificity of 0.855 and 0.964, respectively. In addition, the performances of the radiomic models were validated in the independent cohort. In the test set, the predictive efficacies of the Lesion_D and Δlesion models were 0.875 and 0.837, the sensitivity values were 0.920 and 0.680, and the specificity values were 0.826 and 0.913, respectively. There was no statistical difference in the predictive efficacy between the training set and the test set for all six models (all *p* > 0.05).

In both the training and test sets, the predictive performances of the models based on discharge CTs are higher than those of the models based on admission CTs, and the predictive performances of the models based on lesions are higher than those of the models based on total lung.

## 4. Discussion

Predicting the presence of residual lung lesions in COVID-19 survivors during the recovery period is challenging. In this study of 162 patients, we established six CT-based radiomic models to predict the recovery of residual lung lesions in COVID-19 survivors, at three months after discharge. We found that some of the models (the Δlesion and Lesion_D models) showed good performance in predicting RLL and NRLL, which proved that a CT-based radiomic model is feasible.

In our retrospectively collected cohort of 162 patients, there was no statistical difference in the follow-up interval after discharge from hospital between patients in the RLL and NRLL groups, which ruled out the influence of follow-up time on the recovery of residual lung lesions in COVID-19 patients. In addition, we found that residual lung lesions were more likely to occur in older COVID-19 patients, which was consistent with the findings of Han et al. [16]. Elderly patients have poor physical function and slow recovery; therefore, it was logical to obtain such a result.

A chest CT is significant in the assessment of lung lesions caused by COVID-19 [17,18]. A CT is widely used for admission screening, discharge assessment, and follow-up of COVID-19 patients [7,19,20,21]. Therefore, our radiomic models based on admission and discharge chest CTs of COVID-19 survivors can be well validated. In order to further explore the contribution of CT-based radiomic features in different regions (entire lung or lesion) and at different time points in predicting RLL and NRLL, we constructed six different models and compared them. We found that the predictive efficacies of models based on discharge CTs were better than those of models based on the admission CTs, in both the training and test sets, which may indicate that a discharge CT can provide more valuable information about residual lung lesions than an admission CT. We speculate that this is because the time point of the discharge CT was closer to the follow-up CT than the admission CT. The results also indicate that, in the training and test sets, the predictive efficacies of the models based on lesions are higher than those of the models based on total lung. Moreover, we found that the Δlesion model that combines the admission CT and discharge CT had a better predictive performance than those of the Lesion_A and Lesion_D models, in the training set, and the Δlung model that combines admission CT and discharge CT was worse than those of the Lung_A and Lung_D models alone. This means that lesion-based radiomic features can provide more valuable information than total lung-based radiomic features. The recovery of the residual lung lesions in COVID-19 survivors after discharge from hospital is associated more with the lesions themselves than the state of the total lung. A previous study reported similar conclusions; Wu et al. [22] investigated the CT images of COVID-19 patients during the recovery period and found that the severe group with a higher proportion of ground-glass opacity and consolidation in the total lung required a longer recovery time than the moderate group.

Radiomics refers to the high-throughput extraction and analysis of large amounts of quantitative features from medical images to provide valuable information for the diagnosis, assessment, and prognosis of diseases [23,24,25]. Fang et al. [26] developed and validated a radiomic nomogram to distinguish COVID-19 pneumonia from other types of viral pneumonia. Li et al. [27] extracted radiomic and deep learning features of the lung from CT images to construct a model discriminating critical cases from severe cases of COVID-19. Wu et al. [28] developed a chest CT-based radiomic model in a multicenter cohort to predict a poor outcome of COVID-19 patients (death, mechanical ventilation, or intensive care unit admission). However, to date, no radiomic studies have predicted the recovery of COVID-19 survivors after discharge from hospital. Our study used CT-based radiomic models to predict the presence of residual lung lesions in COVID-19 survivors at three months after discharge. In addition, we constructed a variety of radiomic models based on CT images in different regions and at different time points. We believe our work can provide insight into the lung recovery of COVID-19 survivors.

In this study, each of the six models extracted 1218 candidate radiomic features from CT images. By the mRMR, LASSO, and AIC methods, several potential predictors were finally selected for each model, and these predictors differed significantly between the RLL and NRLL groups. We noticed that there were three radiomic features repeated in the six models, which were log.sigma.3.0.mm.3D_glcm_InverseVariance, wavelet.HLL_glcm_ClusterShade, and log.sigma.3.0.mm.3D_firstorder_Skewness. Log.sigma.3.0.mm.3D_glcm_InverseVariance means a measure of the local dispersion of an image. Wavelet.HLL_glcm_ClusterShade means a measure of the skewness and uniformity of the GLCM. A higher cluster shade implies greater asymmetry about the mean. Log.sigma.3.0.mm.3D_firstorder_Skewness measures the asymmetry of the distribution of values about the mean value. Depending on where the tail is elongated and the mass of the distribution is concentrated, this value can be positive or negative. Wavelet and LoG are both higher-order statistical methods imposing filter grids on the images, and could possibly reflect more information about vascularity and spiculation of a lesion. Wavelets are filter transforms that multiply an image by a matrix of complex linear or radial “waves”. Laplacian transforms of Gaussian bandpass filters can extract areas with increasingly coarse texture patterns from the image.

There are several limitations in our study. First, this study is a retrospective single center study with a small number of cases, and the robustness of our results remains to be validated in the future using data from multiple centers. Second, our study only followed up on the lung recovery of COVID-19 survivors at three months after discharge, while the lung damage caused by COVID-19 requires long-term observation. Third, no other clinical parameters were included in the prediction models. We believe that the next step should be to develop a reliable and robust multi-modality prediction model.

## 5. Conclusions

In conclusion, our results demonstrate the feasibility of a CT-based radiomic model to predict the presence of residual lung lesions in COVID-19 survivors at three months after discharge. This could help doctors to plan follow-up work and reduce the psychological burden of COVID-19 survivors.

## Figures and Tables

**Figure 1 diagnostics-11-01814-f001:**
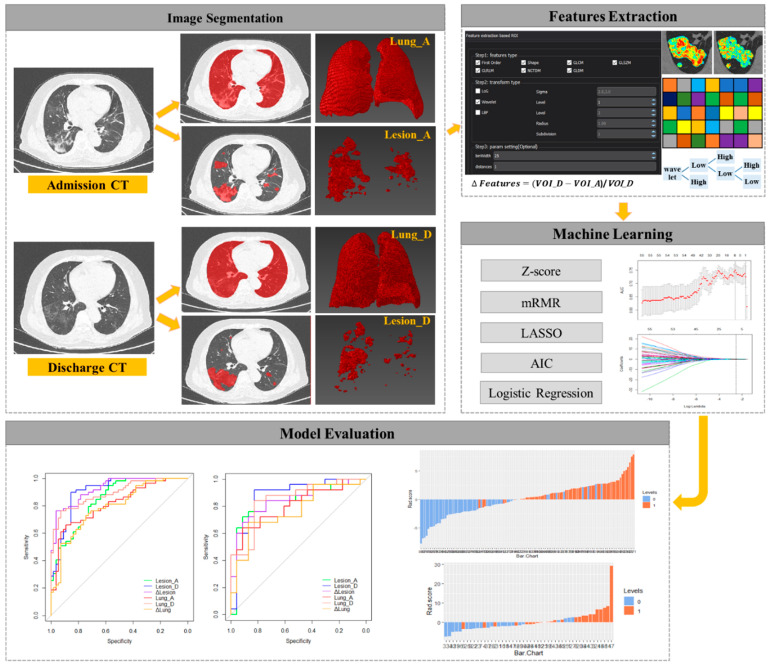
The workflow of the radiomic research.

**Figure 2 diagnostics-11-01814-f002:**
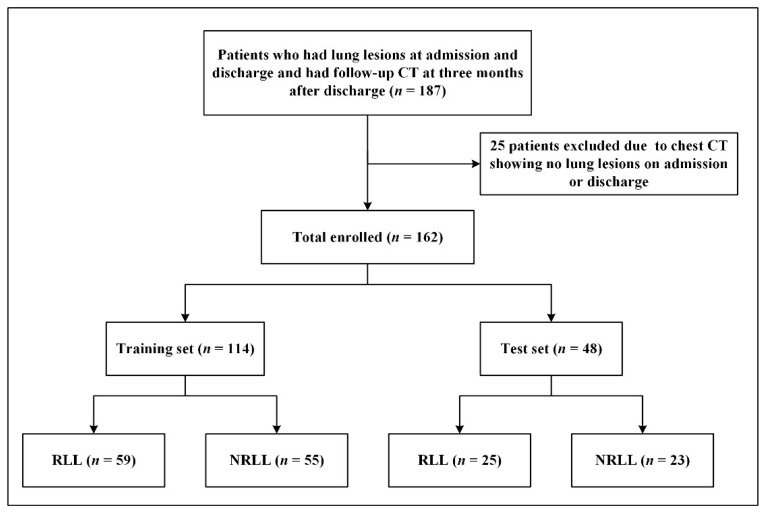
Patient flowchart.

**Figure 3 diagnostics-11-01814-f003:**
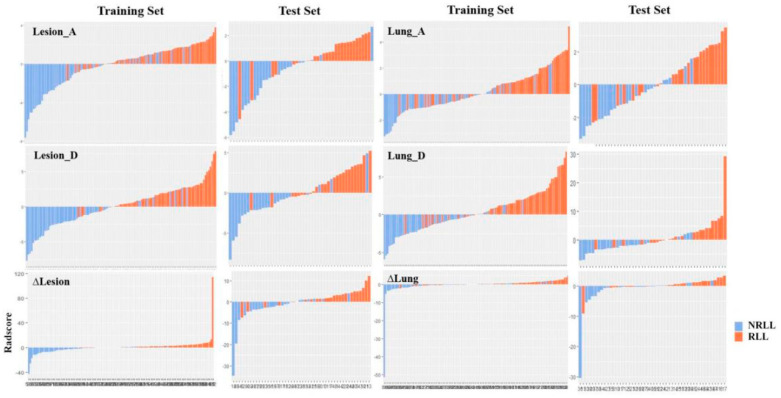
Radiomic scores for each patient in the six models.

**Figure 4 diagnostics-11-01814-f004:**
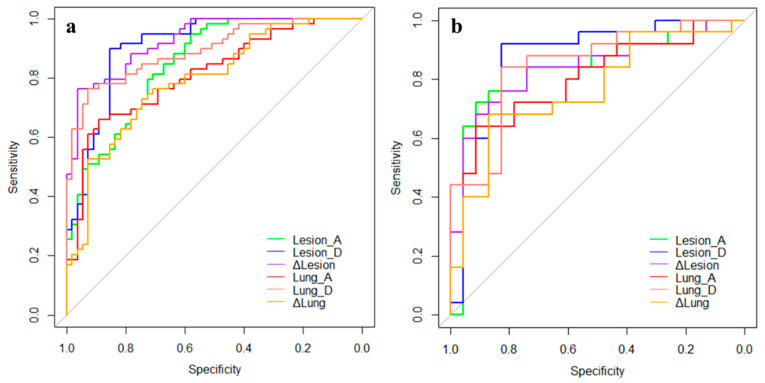
The ROC curves of the six prediction models in the training set (**a**) and the test set (**b**).

**Table 1 diagnostics-11-01814-t001:** Characteristics of patients in the training and test sets.

Characteristics	Training Set (*n* = 114)			Test Set (*n* = 48)		
	RLL (*n* = 59)	NRLL (*n* = 55)	*p*-Value	RLL (*n* = 25)	NRLL (*n* = 23)	*p*-Value
Age (years)	57 (48, 66)	51 (39, 59)	0.005	63 (56, 69)	44 (35, 57)	<0.001
Sex, *n* (%)	-	-	0.071	-	-	0.036
Male	28 (47.5%)	17 (30.9%)	-	14 (56.0%)	6 (26.1%)	-
Female	31 (52.5%)	38 (69.1%)	-	11 (44.0%)	17 (73.9%)	-
Length of hospital stay (days)	20 (16, 27)	18 (11, 28)	0.213	28 (18.5, 36.5)	17 (10, 25)	0.037
Days from discharge to follow-up CT	103 (86, 122)	107 (72, 127)	0.869	94 (79.5, 114.5)	110 (88, 129)	0.137

*p*-values were calculated by Mann–Whitney U test for continuous variables and chi-square test or Fisher’s exact test for categorical variables. RLL patients with residual lung lesions. NRLL patients without residual lung lesions.

**Table 2 diagnostics-11-01814-t002:** Radiomic feature selection results of the six models.

Model	Features	Coefficients
Lesion_A (*n* = 6)	log.sigma.3.0.mm.3D_glcm_InverseVariance	27.3963
	wavelet.HLH_firstorder_Mean	−2.6492
	log.sigma.3.0.mm.3D_firstorder_Skewness	−2.8087
	wavelet.LHH_glcm_Correlation	39.5595
	original_glszm_SmallAreaLowGrayLevelEmphasis	−653.335
	original_shape_Sphericity	−7.1196
Lesion_D (*n* = 8)	log.sigma.3.0.mm.3D_glcm_InverseVariance	40.8774
	wavelet.HLL_glcm_ClusterShade	−0.002
	wavelet.HLL_firstorder_Median	0.2507
	wavelet.HLL_gldm_LargeDependenceLowGrayLevelEmphasis	250.846
	wavelet.LLL_gldm_LargeDependenceHighGrayLevelEmphasis	7.11 × 10^−5^
	original_shape_Maximum2DDiameterRow	0.0153
	log.sigma.4.0.mm.3D_glrlm_LongRunHighGrayLevelEmphasis	0.001
	wavelet.HLH_gldm_SmallDependenceHighGrayLevelEmphasis	0.01845
Δlesion (*n* = 13)	log.sigma.3.0.mm.3D_glcm_InverseVariance	21.7079
	wavelet.HHL_glcm_ClusterShade	0.0154
	log.sigma.3.0.mm.3D_firstorder_Skewness	0.0154
	log.sigma.4.0.mm.3D_glszm_LargeAreaHighGrayLevelEmphasis	−1.7870
	log.sigma.3.0.mm.3D_glszm_SmallAreaEmphasis	10.7100
	log.sigma.3.0.mm.3D_firstorder_Mean	−0.0573
	wavelet.LHH_firstorder_Skewness	0.1581
	log.sigma.4.0.mm.3D_gldm_SmallDependenceLowGrayLevelEmphasis	−0.009
	log.sigma.1.0.mm.3D_firstorder_Mean	0.2537
	log.sigma.1.0.mm.3D_glszm_GrayLevelVariance	6.5212
	wavelet.HHH_firstorder_RootMeanSquared	1551.5109
	log.sigma.1.0.mm.3D_gldm_LargeDependenceLowGrayLevelEmphasis	1.0871
	wavelet.LLL_firstorder_Maximum	−0.0932
Lung_A (*n* = 3)	wavelet.LHL_firstorder_Mean	−0.7123
	wavelet.HHH_glszm_SmallAreaEmphasis	29.0881
	wavelet.LLL_glszm_ZoneEntropy	6.7584
Lung_D (*n* = 7)	wavelet.HHH_firstorder_Mean	39.2045
	wavelet.HLH_firstorder_Skewness	−7.3873
	log.sigma.2.0.mm.3D_glcm_Correlation	−134.446
	wavelet.HHL_glszm_SmallAreaLowGrayLevelEmphasis	7178.18
	wavelet.LHH_firstorder_Median	−18.3834
	wavelet.LLH_glrlm_RunEntropy	9.1189
	wavelet.LLL_glcm_Imc2	21.9867
Δlung (*n* = 5)	wavelet.HHL_firstorder_Mean	−0.1124
	wavelet.LHL_firstorder_Skewness	0.6402
	log.sigma.2.0.mm.3D_glszm_ZoneEntropy	−37.0901
	log.sigma.5.0.mm.3D_gldm_DependenceNonUniformityNormalized	−10.0756
	wavelet.HLH_glszm_LargeAreaHighGrayLevelEmphasis	−0.4602

**Table 3 diagnostics-11-01814-t003:** Comparison of predictive performances of the six models in both the training and test sets.

Model	Training Set			Test Set			*p* Value
	AUC (95% CI)	Sensitivity	Specificity	AUC (95% CI)	Sensitivity	Specificity	
Lesion_A	0.849 (0.780–0.917)	0.949	0.582	0.837 (0.713–0.961)	0.720	0.913	0.8670
Lesion_D	0.907 (0.851–0.962)	0.898	0.855	0.875 (0.766–0.984)	0.920	0.826	0.6115
Δlesion	0.927 (0.883–0.971)	0.763	0.964	0.837 (0.718–0.955)	0.680	0.913	0.1678
Lung_A	0.812 (0.734–0.891)	0.661	0.891	0.809 (0.686–0.932)	0.640	0.913	0.9612
Lung_D	0.893 (0.836–0.950)	0.763	0.927	0.849 (0.738–0.960)	0.840	0.826	0.4890
Δlung	0.791 (0.709–0.873)	0.729	0.745	0.765 (0.628–0.903)	0.680	0.870	0.7490

*p*-values were calculated by using the Delong test to compare the AUCs of the models in the training and test sets.

## Data Availability

The data that support the findings of this study are available from the corresponding author upon reasonable request.

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
