# Peer review of "CT-Based Radiomics Helps to Predict Residual Lung Lesions in COVID-19 Patients at Three Months after Discharge"

_diagnostics, 2021, doi:10.3390/diagnostics11101814_

Round 1

Reviewer 1 Report

The presented work is of high clinical value.

It Deserves publication, after English revision

Either the title deserves improvment to be ellucidative about the obtained Results

Author Response

Response: We appreciate the reviewer’s comments and we have made extensive English revisions to our manuscript. In addition, we have improved the title to make it more concise and accurate.

Reviewer 2 Report

This study by Jia Huang and colleagues investigates the possible basis for using radiomics for evaluating future radiological outcomes in COVID-19 associated pneumonia. The data applies to a hospitalised cohort of patients with radiographic and clinical findings of COVID-19 pneumonia and investigates features on inpatient CT which may predict ongoing radiological abnormalities at 3 months.

I congratulate the authors on their work and agree with their methods and analysis. I have only some minor points of suggestion

Comments

Methods line 65 – please can you clarify what the intended meaning of the sentence is – does this mean informed consent was taken from the participants or was this waived for some reason?

Methods line 88 – how much manual segmentation was required? In all patients or some? Is there a time estimate spent on doing this?

Author Response

1) Methods line 65 – please can you clarify what the intended meaning of the sentence is – does this mean informed consent was taken from the participants or was this waived for some reason?

Response 1: We highly appreciate the reviewer’s comment and we are sorry that the meaning of this sentence is not very clear. Because this is a retrospective study, informed consent was waived. We have revised the text accordingly to make it clear.

2) Methods line 88 – how much manual segmentation was required? In all patients or some? Is there a time estimate spent on doing this?

Response 2: Dear reviewer, we performed manual segmentation in all patients and we have modified the corresponding content of the article to make it clear. Based on the pre-segmentation of uAI Intelligent Assistant Analysis System, it took us about 12 days to perform manual segmentation.